# Cripto Is Targeted by miR-1a-3p in a Mouse Model of Heart Development

**DOI:** 10.3390/ijms241512251

**Published:** 2023-07-31

**Authors:** Tiziana Angrisano, Francesca Varrone, Elvira Ragozzino, Annalisa Fico, Gabriella Minchiotti, Mariarita Brancaccio

**Affiliations:** 1Department of Biology, University of Naples Federico II, 80126 Naples, Italy; 2IRBM S.p.A, 80131 Naples, Italy; fvarrone80@gmail.com; 3Dipartimento di Scienze della Vita e Sanità Pubblica, Università Cattolica del Sacro Cuore, 26100 Rome, Italy; elvira.ragozzino@unicatt.it; 4Stem Cell Fate Laboratory, Institute of Genetics and Biophysics, “A. Buzzati-Traverso”, CNR, 80131 Naples, Italy; annalisa.fico@igb.cnr.it (A.F.); gabriella.minchiotti@igb.cnr.it (G.M.); 5Department of Molecular Medicine and Medical Biotechnology, University of Naples Federico II, 80131 Naples, Italy

**Keywords:** Cripto, miR-1, luciferase assay, cardiac differentiation, cardiac injury

## Abstract

During cardiac differentiation, numerous factors contribute to the development of the heart. Understanding the molecular mechanisms underlying cardiac development will help combat cardiovascular disorders, among the leading causes of morbidity and mortality worldwide. Among the main mechanisms, we indeed find Cripto. Cripto is found in both the syncytiotrophoblast of ampullary pregnancies and the inner cell mass along the primitive streak as the second epithelial–mesenchymal transformation event occurs to form the mesoderm and the developing myocardium. At the same time, it is now known that cardiac signaling pathways are intimately intertwined with the expression of myomiRNAs, including miR-1. This miR-1 is one of the muscle-specific miRs; aberrant expression of miR-1 plays an essential role in cardiac diseases. Given this scenario, our study aimed to evaluate the inverse correlation between Cripto and miR-1 during heart development. We used in vitro models of the heart, represented by embryoid bodies (EBs) and embryonic carcinoma cell lines derived from an embryo-derived teratocarcinoma in mice (P19 cells), respectively. First, through a luciferase assay, we demonstrated that Cripto is a target of miR-1. Following this result, we observed that as the days of differentiation increased, the Cripto gene expression decreased, while the level of miR-1 increased; furthermore, after silencing miR-1 in P19 cells, there was an increase in Cripto expression. Moreover, inducing damage with a cobra cardiotoxin (CTX) in post-differentiation cells, we noted a decreased miR-1 expression and increased Cripto. Finally, in mouse cardiac biopsies, we observed by monitoring gene expression the distribution of Cripto and miR-1 in the right and left ventricles. These results allowed us to detect an inverse correlation between miR-1 and Cripto that could represent a new pharmacological target for identifying new therapies.

## 1. Introduction

Cardiovascular diseases (CVD) are one of the leading causes of death worldwide [1]; therefore, understanding the molecular mechanisms underlying CVDs is of vital importance to preventing premature deaths and curing any CVD-associated pathologies [2].

The heart is one of the first organs to form during embryonic development; it comprises several cell lines, like cardiomyocytes, endothelial cells, epicardial cells, and neural crest cells, which, by interacting with each other, guarantee the correct functioning of the organ [3]. The molecular mechanisms regulating development and morphogenesis include several signaling pathways, such as paracrine interactions, cell–ECM interactions, and cell–cell interaction, which allow the survival, growth, proliferation, differentiation, and migration of the heart tissue [4].

Among the leading agents in cardiac differentiation, it is possible to include the Homeobox protein Nkx 2.5 and transcription factor Gata-4; both are early markers of pre-cardiac cells, making them essential for heart formation [5]. In particular, the cardiac transcription factors Nkx 2–5 and Gata-4 are mutual cofactors that can cooperate transcriptionally to promote the activation of the cardiac-specific atrial natriuretic factor (ANF) [5], which is the primary secretory product of postnatal cardiomyocytes [5]. In addition, it is known that apelin receptor Apj, a class of G protein-coupled receptors, and Myosin light chain-2 (Mlc-2) are involved in cardiac contractility [6,7].

Furthermore, among the cardiac markers, it is necessary to include Troponin [8]. Troponin complex is a component of skeletal and cardiac muscle thin filaments: it consists of three subunits—troponin I, T, and C, and it plays a crucial role in muscle activity, connecting changes in intracellular Ca^2+^ concentration with the generation of contraction [9].

Within the protagonists of cardiomyogenesis, it is also necessary to include Cripto (TDGF1) [10]. Cripto is a member of the EGF–CFC family initially related to epidermal growth factor (EGF) [11]. EGF–CFC proteins have recently been recognized as a new family of extracellular factors required during the early development of vertebrates, in particular, in the formation and correct positioning of the anterior–posterior axis [11]; in addition, the absence of Cripto results in a defective precardiac mesoderm, unable to differentiate into functional cardiomyocytes [12]. Cripto is a glycosylphosphatidylinositol-anchored co-receptor that binds Nodal and the activin type I ActRIB (ALK)-4 receptor (ALK4) [10,11,12,13,14]. Its family includes monkey Cripto-1, mouse Cripto-1 (Cr-1 = cfc2), chicken Cripto-1, one-eyed zebrafish pinhead (oep), Xenopus XCR1/FRL-1, XCR2, and XCR3, mouse cryptic (Cfc1), and human Cryptic (CFC1) [15].

Numerous scientific pieces of evidence have shown that Cripto can be considered a marker of undifferentiated embryonic stem cells in vitro, as it appears to be involved in maintaining pluripotency and self-renewal in both human and mouse stem cells, together with Oct-4 and Nanog [16]. Furthermore, recent findings show that Cripto is expressed during muscle regeneration by inducing the proliferation and migration of muscle stem cells (satellite cells), thus providing evidence that Cripto is a regulator of muscle and satellite cell regeneration (muscle stem cells), directing these towards a myogenic fate [17,18]. More specifically, these results suggest an involvement of Cripto in cardiac and skeletal muscle regeneration [17,18].

Recent scientific evidence has shown that the molecular mechanisms underlying cardiomyogenesis are regulated with a vast collection of microRNAs (miR) that modulate cardiac muscle development and the appearance of any pathologies [19,20]. MiRs are small, endogenous, single-stranded, non-coding RNA molecules found in the transcriptome of plants, animals, and some viruses. They are polymers encoded by eukaryotic nuclear DNA about 20–22 nucleotides long, mainly active in regulating gene expression at the transcriptional and post-transcriptional levels [21,22,23]. MiR-1 is the myomiR most involved in cardiac development in humans and mice [24,25]. In humans, the genes encoding myomiRs are organised into three cistrons encoding (miR-1-2, miR-133a-1), (miR-1-1, miR-133a-2), and (miR-133b, miR-206) and are located on chromosomes 18q11.2, 20q13.33, and 6p12.2, respectively. The two mature miR-1 isomers, as do the two miR-133a isomers, show an identical sequence. In contrast, the mature miR-133 isomers differ only at the 3′ terminal base, with miR-133a1/2 ending in G-3′ and miR-133b ending in A-3′, respectively [26].

Moreover, both miR-1/-133a gene clusters are canonically expressed in skeletal and cardiac muscle. The miR-133b/-206 gene cluster is expressed in developing skeletal muscle but not in cardiac muscle, defining the roles of myomiRs in muscle biogenesis [26].

Studies of individual miRs using developmental models of the heart have discovered that miR-1 is fundamental to controlling proliferation and regulating transcriptional muscle networks [19]. Ivey et al. have shown that miR-1 regulates the fate of cell lines in mouse and human embryonic stem cells (ES); in fact, it promotes mesoderm formation from ES cells. Still, it has opposing functions on endodermal and ectodermal precursors [27].

Moreover, miRNAs interact with target sites in the 3′ untranslated regions (3′UTR) to regulate mRNA expression [28]. Recent studies have shown that miR-1 binds Histone deacetylase 4 (*HDAC-4*) to the *3′UTR* and represses its expression, promoting differentiation into cardiomyocytes in human progenitor cells [29]. Therefore, in vitro systems suitable for studying and discerning the molecular mechanisms underlying cardiomyogenesis represent useful scientific tools for identifying new targets and biological processes. For example, embryonic bodies (EB) represent an adequate in vitro system capable of reproducing the interactions that occur during normal gastrulation and allow for the specification of primary germ layers, including the mesoderm from which cardiomyocytes are derived [30,31,32].

In this scenario, our study aimed to develop an in vitro heart system to validate and demonstrate the cross-talk between Cripto and miR-1. To detect the cross-talking between Cripto and miR-1, we carried out: (I) bioinformatic analysis; (II) luciferase assay; (III) evaluation of Cripto and miR-1 expression in EBs; (IV) monitoring of the gene expression levels of key components of cardiomyogenesis (*Nkx 2.5*, *Gata-4*, *Apj*, and *Mlc*-2) during cardiac differentiation of the mouse teratocarcinoma-derived cell line (P19 cells) and, at the same time, Western blot analysis of cardiac Troponin T; (V) detection of *Cripto* and miR-1 expression during cardiac differentiation of P19 cells; (VI) gene silencing experiments and analysis of the correlation between *Cripto* and miR-1 in P19 cells before and after cardiac differentiation in P19 cells; (VII) assessment of *Cripto* and miR-1 gene expression after cardiac differentiation in P19 cells subjected to injury with cobra cardiotoxin; (VIII) demonstrating the inverse correlation between Cripto and miR-1 using Pearson’s correlation as a statistical test; (IX) evaluation of Cripto and miR-1 gene expression in adult mouse heart biopsies.

## 2. Results

### 2.1. Complementarity Prediction between miRNA-1 and Cripto

To verify if *Cripto* was targeted by miR-1, we performed a prediction by TargetScan: an online analysis tool [33,34]. Bioinformatic analysis revealed that Mmu-Cripto (ENSMUST00000035075.12) is a target of mmu-miR-1a-3p (miR-1) (Figure 1). 

To obtain a detailed picture of the sequence similarity between Hsa-miR-1-3p (MIMAT0000416) and Mmu-miR-1a-3p (MIMAT0000123), we performed a Blastn alignment, and, in our case, we found a sequence identity equal to 100%; the same approach was used for *Mmu-Cripto* (NM_011562.2) and *Hsa-Cripto* (NM_003212.4), and, in this case, we found an identity equal to 80%.

Despite the high sequence similarity between Hsa-miR-1-3p and Mmu-miR-1a-3p and between Hsa-Cripto and Mmu-Cripto in prediction databases such as TargetScan, in humans, we do not find that Cripto is a target of miR-1-3p, while for the rat (ENSRNOT00000040840.6), the 3′UTR of the gene is not annotated in databases. 

To experimentally establish this gene as a target of miR-1, we subcloned the 3′UTRs of *Cripto* into the 3′UTR of a luciferase plasmid to construct chimeric vectors. Co-transfection of the chimeric vectors with miR-1, in HEK-293 cells, resulted in lower luciferase activity than the transfection of chimeric vectors alone (Figure 2). The repression of *Cripto* luciferase activity by miR-1 was alleviated by co-transfection of AntagomiR-1 (Figure 2A). The scramble (negative control) sequences produced no effects on the luciferase activity of the chimeric vectors (Figure 2A). These results indicate that *Cripto* is a direct target of miR-1. Data were confirmed by protein assay by Western blot (Figure 2B,C). 

### 2.2. Expression Level of miRNA-1 and Cripto in Mouse EBs during Cardiac Differentiation

To demonstrate the inverse relationship between miRNA-1 and Cripto during cardiac differentiation, we differentiated EBs into cardiomyocytes, as previously described in Minchiotti et al. [35] and schematically indicated in Figure 3A. We then performed a qPCR to evaluate the expression levels of miRNA-1 and *Cripto* (Figure 3B). In this case, we highlighted a significant increase in Cripto on day four that then disappeared (Figure 3B black bars), while for miRNA-1, an opposite trend was observed (Figure 3B); we have a peak at day 8 (Figure 3B). Data were confirmed through Western blot assay (Figure 3C).

### 2.3. Evaluation of Gene Expression of the Leading Agents of Cardiomyogenesis in P19 Cells Undergoing Cardiac Differentiation

First of all, we differentiated P19 cells into cardiomyocytes, as shown in Figure 4A; in particular: we plated the cells at a density equal to 5 × 10^5^ cells/well, and after 6 h, we treated the cells with 10 µM of Azacytidine (5-Aza), a cytosine analogue capable of altering expression of specific genes that may regulate differentiation [36,37], for 24 h. Finally, we exposed the cells to DM (containing α-MEM supplemented with 2% dimethyl sulfoxide) [38] (Figure 4A). To verify that the cardiac differentiation induced in P19 cells, as described in Figure 4A, was efficient, we evaluated the protein expression of cardiac Troponin T (Troponin T) by Western blot analyses (Figure 4B,C). The data in our possession show that as the days of differentiation increase, the protein level of Troponin T also increases (Figure 4B); this result agrees with what was previously described [8,9,39,40].

Therefore, to evaluate whether the cardiac differentiation induced in P19 cells (Figure 4A) was adequate, we assessed the gene expression levels of the main protagonists of cardiomyogenesis (Figure 4D–G). As shown in Figure 4B,C, *Nkx 2.5* and *Gata-4* increase in the first days of differentiation and then decrease; the opposite trend is observed for *Apj* and *Mlc-2,* which increase, respectively, around day two and day four (Figure 4F,G).

Following that, since bioinformatic analysis highlighted that Cripto is a target of miR-1, we wanted to understand if the regulation occurred only on the 3′UTR or on the entire Cripto sequence; we monitored the expression levels of *Cripto, Cripto-UTR, Hdac-4,* and miRNA-1 (see Figure 5A–C). The gene expression levels of *Cripto, Cripto-UTR,* and *Hdac-4* decrease with increasing days of differentiation (Figure 5A,B), while miR-1 increases with successive days of differentiation (Figure 5C).

### 2.4. Silencing the miR-1 Gene

To reveal if miR-1 had a positive effect on *Cripto*, we carried out a transfection of AntagomiR-1 into P19 cells (Figure 6A). The first two steps of the transfection protocol are identical to schematic Figure 4A, after which the cells are exposed to 4h of α-MEM alone and finally transfected for 24 h (Figure 6A).

After that, we evaluated the gene expression levels of *Cripto, Cripto-UTR*, and miR-1 after 24 h of transfection (Figure 6B). As can be seen from the graph shown in Figure 6B, the miR-1 silencing implies an increase in the levels of *Cripto* and *Cripto-UTR* (Figure 6B), if compared to the non-transfected cells; the opposite trend is noted of miR-1 (Figure 6B).

In addition, we performed a second transfection experiment in which, after 24 h of gene silencing, the cells were differentiated for 10 days (see scheme in Figure 7A), first using the transfection protocol and then the differentiation protocol.

In this case, we verified whether the 24 h transfection affected the differentiation and interfered with the expression of *Cripto, Cripto-UTR*, and miR-1.

From the graph shown in Figure 6B, we can observe that on day 6 of differentiation, there is a switch between *Cripto, Cripto-UTR*, and miR-1. In fact, at day 6, the levels of *Cripto* and *Cripto-UTR* start to decrease, and those of miR-1 increase, if compared to the non-transfected (Figure 7B).

### 2.5. Monitoring the Relationship between miR-1 and Cripto during CTX Damage

To shed light on the role of miR-1 and *Cripto* during cardiac injury, we evaluated the levels of their expression inside (in cells) and outside (in culture medium) of differentiated P19 cells in cardiomyocytes for six days and then treated them for 24 h with a CTX 1 µM solution (see schematic Figure 8A). We decided to use CTX, as it induces perturbations of cytosolic calcium homeostasis and hypercontraction in adult rat ventricular myocytes [41]; in contrast, miR-1 is known to regulate calcium signaling during heart disease [42].

From the graph shown in Figure 7B, we highlight that after six days of differentiation, the levels of *Cripto* and *Cripto-UTR* decrease compared to day 0, while the miR-1 levels increase. In contrast, after damage with CTX, we observe that inside the cell, the levels of *Cripto* increase, while those of miR-1 decrease (Figure 8B).

In addition, in the graph shown in Figure 8C, in which we evaluated the levels of *Cripto, Cripto-UTR*, and miR-1 in the culture medium to understand trafficking, the same trend is highlighted post-differentiation (Figure 8C); however, after damage, we can note an increase of miR-1 in the medium and a simultaneous decrease of *Cripto* and *Cripto-UTR* when compared to undamaged cells (Figure 8C).

### 2.6. Correlation between Cripto and miR-1 

To shed light on a possible correlation between the gene expression values of Cripto and miR-1 obtained by differentiating P19 cells, we used Pearson’s linear correlation coefficient [43] (Table 1). From the data obtained, we can state that there is an inverse correlation between Cripto and miR-1 from day 6; however, this correlation was weak on day 6, modest on day 8, and strong on day 10 (see Table 1 and Figure 9), thus suggesting a possible interaction between Cripto and miR-1 during cardiac development.

### 2.7. Evaluation of Cripto and miR-1 in Adult Mouse Heart Biopsies

Finally, to understand the distribution of miR-1 and *Cripto* at the ventricular level, we measured their expression levels in ex vivo tissues of the hearts of adult mice (Figure 10). In this case, we found that the Cripto levels are almost identical between the two ventricles (Figure 10), while there is a slight but significant increase of miR-1 at the level of the left ventricle (Figure 10).

## 3. Discussion

In recent years, research has focused on detecting appropriate markers for the early identification of cardiovascular disorders, emphasising cardiac pathologies [44]. The molecular mechanisms underlying cardiomyogenesis and possible cardiac regeneration pathways remain fundamental for identifying early biomarkers and new targets for targeted therapies [45,46,47]. At the same time, the use of bioinformatic approaches for the search for new therapeutic targets is now essential for the identification of new molecular targets; however, validation is still a process that requires understanding the role of the gene or protein in the disease process [48]. 

Among the predicted targets of miR-1 identified by computational analysis, Cripto is involved in cardiomyogenesis.

Previous studies by Parisi et al. showed that Cripto^−/−^ ES cells treated with a recombinant Cripto protein caused increased Smad2 phosphorylation, suggesting that Cripto signaling acts via the Smad2 pathway to promote cardiac induction and reveals a potential role of Nodal signaling in cardiogenesis [10]. This activation was, however, insufficient to obtain a correct terminal cardiac differentiation [10]. 

In our case, in EBs, we highlighted an increase in Cripto around day four that then disappeared on day eight: this expression profile of Cripto is in line with that previously shown by Parisi et al. [10]. On the other hand, the opposite trend was observed for miR-1. These data suggested that during the first phases of cardiac differentiation in EBs, the main agent was Cripto; however, in the advanced stages of differentiation, the regulator was miR-1 [49]. Cardiac myocytes are known to derive from the embryonic mesoderm during development. The transcriptional network that regulates cardiomyogenesis involves serum response factor (SRF) and myocyte enhancer factor 2 (MEF2). Recently, SRF and MEF2 were found to regulate the expression of two sets of muscle-specific miRNA (myomiRNAs) genes: miR-1-1/miR133a-2 and miR1-2/miR133a-1. Two other myomiRNAs are miR-208 and miR-206, both involved in cardiac contractility and skeletal muscle development [49,50]. Emerging evidence demonstrates that a complex network of myomiR-post-transcriptional regulated gene expression coordinates overall cardiomyocyte development and function [46]; consequently, the result that high levels of Cripto gene expression coincide with low levels of miR-1 and vice versa is justified by the role of these two key agents in cardiac differentiation separately [49,50,51].

To support the results obtained in ESC cardiac differentiation, we used teratocarcinoma-derived P19 cells [52], and, to verify the proper differentiation of P19 cells into cardiomyocytes, we evaluated the gene expression of key markers of cardiac differentiation: *Nkx 2.5, Gata-4, Apj*, and *Mlc-2*. The results obtained agree with the literature, i.e., in fact, *Nkx 2.5* and *Gata-4* are expressed in the first stages of differentiation [5]. Their down-regulation allows us to affirm that the cells are differentiating correctly and that there are no exogenous or endogenous perturbations; in fact, it is known that their re-activation can be caused by an activation of MAPKs due to an accumulation of ROS or external perturbations [53,54]. Furthermore, we highlight a two- to four-day increase in *Apj*, which is known to be co-expressed with Cripto in cardiac ESC differentiation [55]. Finally, *Mlc-2* appears later, being involved in contractility [7]. Simultaneously, we evaluated the expression of *Cripto, Cripto-UTR, Hdac-4*, and miR-1: we assessed both *Cripto* and only *Cripto-UTR* because, from our bioinformatic analysis, miR-1 bound to the *3′UTR Cripto*. Furthermore, we decided to monitor the expression of *Hadc-4*, being a known target of miR-1 [28]: miR-1 binds *Hadc-4* to the *3′UTR* and represses its expression, promoting myogenesis [28], and probably, in a similar molecular mechanism, when miR-1 binds to the *3′UTR Cripto*, it suppresses its expression and announces cardiomyocyte specification.

In addition, we analysed the protein increase of Troponin T; Jasmin Spray et al. have shown that the chemical differentiation induced in P19 cells using DMSO is efficient, as there is an increase of Troponin T with the increase of days of differentiation [39]; our data confirm what is seen previously in the literature [38]. These results substantiate again how P19 cells are a suitable in vitro model to study the molecular mechanisms of cardiac differentiation, as already demonstrated in van der Heydena et al. [40].

To confirm these results, we decided to silence miR-1 and to evaluate whether this silencing interfered with *Cripto* expression. Specifically, cells subjected to miR-1 gene silencing expressed high levels of *Cripto* compared to non-transfected cells. This experiment suggests that there is cross-talk between miR-1 and *Cripto* during cardiomyogenesis. In particular, in the first gene silencing experiment, the cells are subjected exclusively to a 5-Aza treatment (a wave of demethylation), which is known to favour cardiac differentiation [36,37]; in the second experiment instead, we first spent the miR-1 and then subjected the cells to cardiac differentiation through the use of DMSO [39]; in this case, it is highlighted that the switch between Cripto and miR-1 starts around day 6 instead of day 4 as highlighted only in the differentiation (see Figure 5 and Figure 7, respectively). Consequently, miR-1 silencing re-validates the hypothesis that Cripto and miR-1 are “related” during cardiac differentiation, as demonstrated by Pearson’s linear correlation from our data. 

In addition, to confirm cross-talk during cardiomyogenesis, we damaged differentiated cells with CTX to mimic cardiac injury. Then, we evaluated gene expression levels of *Cripto* and miR-1 inside and outside the cell. Our data show that during CTX-induced damage, *Cripto* increases inside and decreases outside the cell, while miR-1 has an opposite trend.

These results highlight two fundamental aspects: first, during cardiac damage, the cells break down and release the miRNAs into the extracellular space; secondly, factors involved in self-renewal are re-expressed during heart damage, as previously seen in [19,54,55]. Consequently, miR-1 could be used as an early damage biomarker since it is released and would be easily measurable [56,57]. On the other hand, *Cripto*, being involved in the maintenance of stemness [53], could be used for a possible tissue regeneration process.

Finally, we detected the gene expression of Cripto and miR-1 in adult mouse biopsies, highlighting that the expression of *Cripto* is equally distributed between the right and left ventricles; at the same time, miR-1 is mainly expressed in the left ventricle, as reported in the literature [58,59]. 

Our data show the fundamental role of miRNAs in the regulation of post-translational processes [60,61,62,63]; in particular, we highlight a positive effect of miRNA-1 on *Cripto,* which appears to play a crucial role in cardiomyogenesis. At the same time, the results, in which we used the whole heart tissue, allow us to consider the possibility of exploiting miR-1 and Cripto as cardiac markers [64].

## 4. Materials and Methods

### 4.1. In Silico Analysis

To verify that Cripto was a target of miR-1, we employed TargetScan 7.2, an online database [33,34].

In addition, to get a complete picture of the sequence similarity between Human-miR-1-3p [Hsa-miR-1-3p (Has-miR-1); MIMAT0000416] and Mouse-miR-1a-3p [Mmu-miR-1a-3p (Mmu-miR-1); MIMAT0000123], we performed a Blastn [65]. The same procedure was applied to Human-Cripto (Hsa-Cripto; NM_003212.4) and Mouse-Cripto (Mmu-Cripto; NM_011562.2). 

### 4.2. Synthesis of miR-1 and Anti-miR-1 Antisense Inhibitor (Antagomir-1)

In this study, miR-1 and its antisense oligonucleotides Antagomir-1 were synthesised by CEINGE—Oligo Synthesis Services (Naples, Italy). Additionally, a scrambled RNA was used as a negative control (NC); miR-1, sense: 5′-UGGAAUGUAAAGAAGUGUGUAU-3′ and antisense: 5′-AUACACACUUCUUUACAUUCCA-3′. All pyrimidine nucleotides in the NC or miR-1 were substituted by their 2′-O-methyl analogues to improve RNA stability [66].

### 4.3. Luciferase Assay

To construct reporter vectors bearing miRNA-target sites, we first obtained fragments of the 3′UTRs of *Cripto* containing the exact target sites for miR-1 by PCR amplification; then, 3′UTR fragments were inserted into the multiple cloning sites downstream the luciferase gene (HindIII and SacI sites) in the pMIR-REPORTTM luciferase miRNA expression reporter vector (Ambion, Inc., Austin, TX, USA) to form chimeric plasmid.

After that, 1 µg of the chimeric plasmid (firefly luciferase vector), 0.1 µg PRL-TK (TK-driven Renilla luciferase expression vector), and the appropriate miRNA or AntagomiR were co-transfected with Lipofectamine 2000 (Invitrogen, Waltham, MA, USA) into human embryonic kidney 293 cells (HEK-293 cells) (1 × 10^5^ cells/well) [66]. Luciferase activities were measured with a dual luciferase reporter assay kit (Promega, Madison, WI, USA) on a luminometer (GloMax™ 20/20, Madison, WI, USA) 48h following transfection [66]. For all experiments, transfection took place 24 h after starvation of cells in a serum-free medium [66]. The normalised luciferase activity relative to the control group was used to demonstrate the alteration of Cripto activity [66]. 

HEK-293 is a cell line that was isolated from the kidney of a human embryo; Ishii et al. have used this cell line as a negative control of the expression of Cripto in the formulation of a new, artificially humanized anti-Cripto-1 antibody that suppresses the growth of tumour cells [67]. It is known that Cripto is not expressed in the kidney, as reported in the Protein atlas database (https://www.proteinatlas.org/ENSG00000136698-CFC1/tissue, accessed on 1 June 2023), so we used these cells to perform a luciferase assay to monitor exogenous Cripto levels in the absence and presence of miR-1.

### 4.4. ES Differentiation

Undifferentiated ES cells were cultured as previously described [68]. For in vitro differentiation, ES cells were cultivated in Ebs, essentially as previously described [35]. The Ebs were plated separately onto gelatin-coated 100 mm tissue culture plates for RT-PCR. 

### 4.5. Cell Cultures 

Human HEK-293 cells were obtained from ATCC and cultured in the following media: Dulbecco’s Modified Eagle Medium (DMEM), supplemented with 10% Fetal Bovine Serum (FBS; Gibco), 1% glutamine, and 1% antibiotics (100 U/mL penicillin and 100 μg/mL streptomycin; Gibco).

Mouse P19 cells were obtained from ATCC and cultured in the following media: Alpha Minimum Essential Medium with ribonucleosides and deoxyribonucleosides (α-MEM), supplemented with 7.5% Fetal Bovine Serum (FBS; Gibco), 1% glutamine, and 1% antibiotics (100 U/mL penicillin and 100 μg/mL streptomycin; Gibco); Differentiation Medium (DM) containing α-MEM supplemented with 2% dimethyl sulfoxide (DMSO, Life Technologies). P19 cells were seeded in 6-well plates (5 × 10^5^ cells/well), and before inducing cardiac differentiation using DM, the cells were pre-treated with 10 µM of 5-Azacytidine (5-Aza, Life Technologies) for 24 h [36,37]. The DM was changed every 48 h to increase the efficiency of differentiation [38,39,40].

### 4.6. AntagomiR-1 Transfection

P19 cells were seeded in 6-well plate format (2.5 × 10^5^ cells/well) in α-MEM for 6 h and then pre-treated with 10 µM of 5-Aza for 24 h. After that, the cells were treated for 4 h with α-MEM and transfected with 50 nM of AntagomiR-1 and scramble as a negative control (Exiqon, Copenhagen, Denmark), using the Lipofectamine 2000 (Invitrogen) reagent according to the manufacturer’s protocol.

The same experiment was performed in cells that, after 24 h of transfection, were treated with DM for 10 days.

### 4.7. Cardiomyocyte Treatments and Analysis

To assess miR-1 and Cripto expression during injury, P19 cells were seeded in a 6-well plate format (5 × 10^5^ cells/well) in α-MEM for 6 h and then pre-treated with 10 µM of 5-Aza for 24 h. After that, the cells were treated for 6 days with DM, and then the damage was performed by 1 μM Cardiotoxin (CTX) (Sigma, Saint Louis, MO, USA) treatment for 24 h.

### 4.8. Ex Vivo Mouse Heart Tissues

C57Bl/6 J mice were used in this study. Adult mice were sacrificed by cervical dislocation, and the hearts were removed. Hearts explanted from a group of animals (6 mice) were dissected to obtain left and right ventricles. Before being suitably frozen at −80 °C, the tissue was washed in 1X phosphate buffered saline (PBS, Life Technologies, Carlsbad, CA, USA). All animal studies were performed under approved protocols by the Institute of Genetics and Biophysics, ‘A. Buzzati-Traverso’, CNR and were conducted according to EU Directive 2010/63/EU for animal experiments.

### 4.9. RNA Extraction and cDNA Synthesis

Total RNA was extracted from cultured cells and heart tissues using Trizol Reagent according to the manufacturer’s protocol (Life Technologies). The amount of total extracted RNA was estimated by measuring the absorbance at 260 nm and the purity in 260/280 and 260/230 nm ratios by Nanodrop (ND-1000 UV–Vis Spectrophotometer, NanoDrop Technologies, Wilmington, DE, USA). For each sample, 1000 ng of total RNA was retrotranscribed using a High-Capacity cDNA Reverse Transcription kit (Applied Biosystems, Foster City, CA, USA) according to the manufacturer’s protocol. 

### 4.10. Gene Expression by Real-Time qPCR

The data from each cDNA sample were normalised for real-time qPCR experiments using the mouse housekeeping gene *Gapdh* (glyceraldehyde 3-phosphate dehydrogenase). The specific primers used for amplification of *Gapdh* (NM_001289726.2), *C*ripto-UTR (NM_011562.2), *Cripto* (NM_011562.2), *Nkx 2.5* (NM_008700.2), *Gata-4* (NM_001310610.1), *Apj* (NM_011784.3), and *Mlc-2* (NM_010861.4) were designed based on the nucleotide sequences downloaded from the NCBI database using Primer3WEB v.4.0.0 (see Table 2). For microRNA quantitative reverse transcription–polymerase chain reaction, primers for mature miR-1 and the internal control U6 were used according to the manufacturer’s protocol (MiRCURY LNA microRNA system control primer set, Exiqon) [69]. 

Calculations of relative expression levels were performed using the 2^−ΔΔCt^ method [70,71]. All analysis was performed in triplicate to guarantee the accuracy of results.

### 4.11. Protein Extraction 

To obtain total protein extracts, cells were washed with cold PBS and resuspended in RIPA lysis buffer (1×) containing 50 mM Tris-HCl (pH 7.6), 150 mM NaCl, 5 mM EDTA, 0.5% NP-40, 0.5% sodium deoxycholate, 10% SDS, phosphatase, and protease inhibitor cocktail (Roche, Basel, Switzerland). Cell homogenates were centrifuged at 13,000× *g* for 5 min at 4 °C, and the supernatant was used as a total protein extract. The protein content of the total extracts was determined with the Bio-Rad protein assay reagent using bovine serum albumin as standard.

### 4.12. Western Blot Analysis

Total protein extracts were analysed on 12% SDS/polyacrylamide using Laemmli buffer. Following electrophoresis, proteins were transferred onto a PVDF (Millipore, Burlington, MA, USA) membrane (Bio-Rad Trans-Blot Apparatus) and probed with rabbit polyclonal antibodies: anti-Cardiac Troponin T (from Abcam catalogue number: ab115134), diluted 1:1000 in non-fat dried milk 5% in PBS or with rabbit polyclonal to Cripto1/CRIPTO (from Abcam catalogue number: ab19917), diluted 1:1000 in non-fat dried milk 5% in PBS and, as an internal control, the mouse anti-Gapdh monoclonal antibody (from Abcam catalogue number: ab8245) diluted 1:1000 in PBS milk 5%. The appropriate secondary anti-mouse and anti-rabbit HRP-conjugated antibodies (Amersham), both diluted 1:10,000 in milk 5%, were added at 37 °C for 1 h, and immune-reactive proteins were detected using the ECL (WesternBrightTM detection kit ECL, Advansta, San Jose, CA, USA) according to the manufacturer’s instructions. Immunopositive bands were analysed by densitometry using the Image J software v.1.53t. 

### 4.13. Statistical Analyses

All statistical analyses were performed using GraphPad Prism 8.0.1 (GraphPad Software Inc., La Jolla, CA, USA). All data are the results of at least three independent experiments carried out in triplicate. Data were expressed as the means and standard deviations (SD) [72]. As appropriate, comparisons among groups were made by Student’s *t*-test or analysis of variance ANOVA, followed by Dunnett’s multiple comparison test. Values of *p* < 0.05 were considered significant. To evaluate the relationships between Cripto and miR-1, Pearson’s linear correlation coefficient was used [43]. Here, a value of +1 corresponded to a perfect positive linear correlation; 0 corresponded to an absence of a linear correlation; and −1 corresponded to a perfect negative linear correlation [43]. 

## 5. Conclusions

In conclusion, the results obtained from our study highlight, on the one hand, how miR-1a-3p and *Cripto* can be used as cardiac biomarkers, and, on the other hand, the possibility of using both as new therapeutic targets of personalised therapies in case of heart damage.

Obviously, further studies using models physiologically closer to in vivo, such as cardiac organoids, will be needed, which will help us to clarify the importance of miR-1 and Cripto in post-damage regeneration.

## Figures and Tables

**Figure 1 ijms-24-12251-f001:**
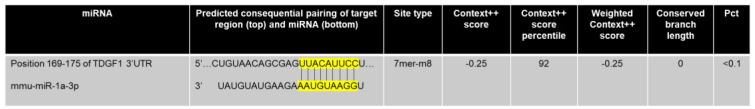
Bioinformatic prediction. The overlapping target gene was predicted through TargetScan. Context++ score and features that contribute to the context++ score are evaluated as in Agarwal et al. [33]. Conserved branch lengths and P_CT_ are evaluated as in Friedman et al. [34], with an expanded 46-species alignment as described in Agarwal et al. [33].

**Figure 2 ijms-24-12251-f002:**
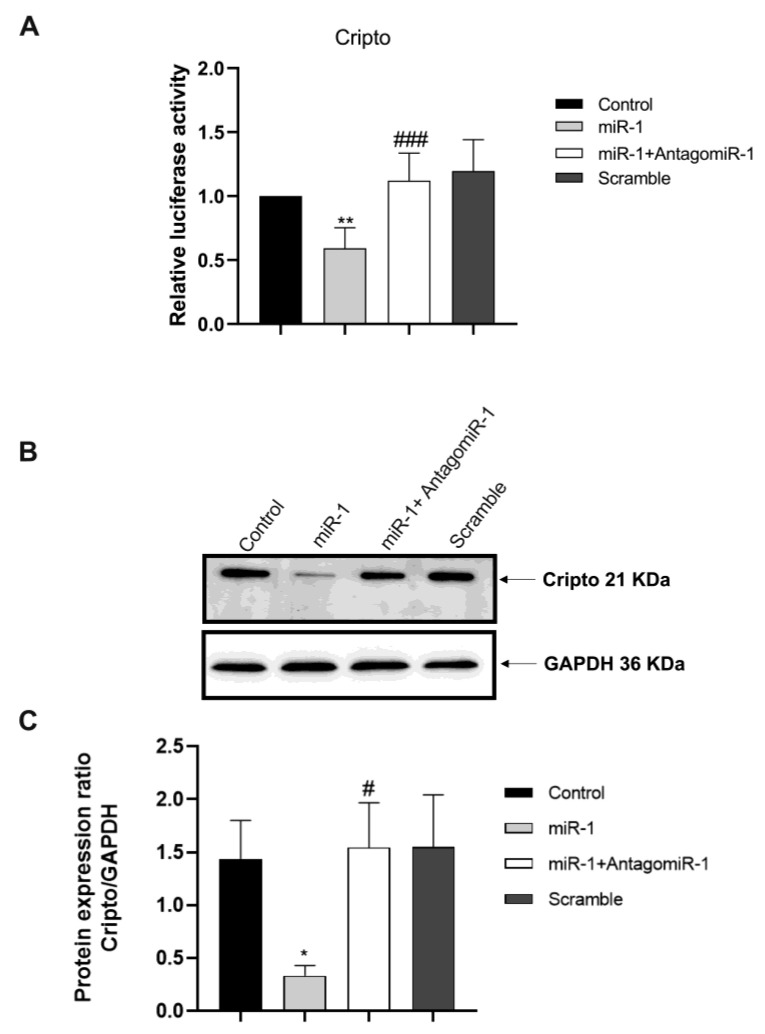
*Cripto* is a target of miR-1. (**A**) Luciferase reporter activities of chimeric vectors carrying the luciferase gene and a fragment of *Cripto* 3′UTR containing the binding sites miR-1 and co-transfection with AntagomiR-1. (**B**,**C**) Cripto protein expression analysis after miR-1 transfection and Antagomir co-transfection. The data are expressed as the means (SDs). The significance was determined by one-way ANOVA followed by Dunnett’s multiple comparison test. (**A**) ** (*p* < 0.01) vs. Control and ^###^ (*p* < 0.001) vs. miR-1. (**C**) * (*p* < 0.05) vs. Control and ^#^ (*p* < 0.05) vs. miR-1.

**Figure 3 ijms-24-12251-f003:**
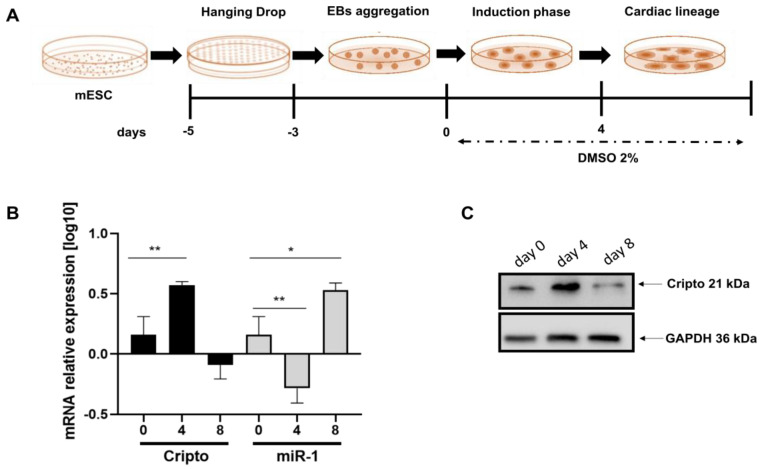
Gene expression of miRNA-1 and Cripto in mouse EBs. (**A**) Schematic representation of the experimental protocol used for ES cell differentiation into cardiomyocytes. (**B**) qPCR of miRNA-1 and Cripto gene expression in EBs during cardiomyogenesis. The mRNA levels of Cripto were normalised to *Gapdh* levels, while the miR-1 levels were normalised to U6 expression. The data are expressed as the means (SDs). The significance was determined by one-way ANOVA, followed by Dunnett’s multiple comparison test; * (*p* < 0.05) and ** (*p* < 0.01). (**C**) Representative Western blot performed on mouse EBs using Cripto-specific antibody.

**Figure 4 ijms-24-12251-f004:**
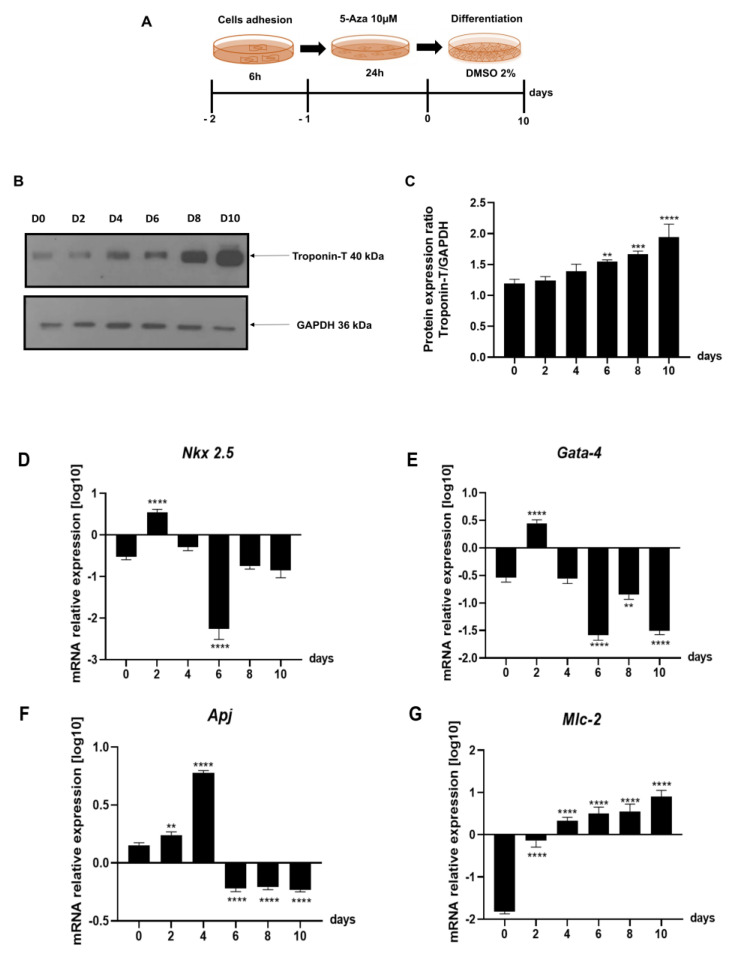
Cardiomyogenesis markers in in vitro model of cardiac differentiation. (**A**) Schematic representation of the experimental protocol used for P19 cell differentiation into cardiomyocytes and (**B**,**C**) Troponin T protein expression analysis in P19 cells during cardiac differentiation by Western blot and densitometric analysis with ImageJ software, respectively. The protein levels of Troponin T were normalised to Gapdh protein levels. (**D**–**G**) qPCR of cardiomyogenesis markers in P19 cells during cardiac differentiation. The mRNA levels were normalised to *Gapdh* levels. All data are expressed as the means (SDs). The significance was determined by one-way ANOVA followed by Dunnett’s multiple comparison test; ** (*p* < 0.01), *** (*p* < 0.001), and **** (*p* < 0.0001).

**Figure 5 ijms-24-12251-f005:**
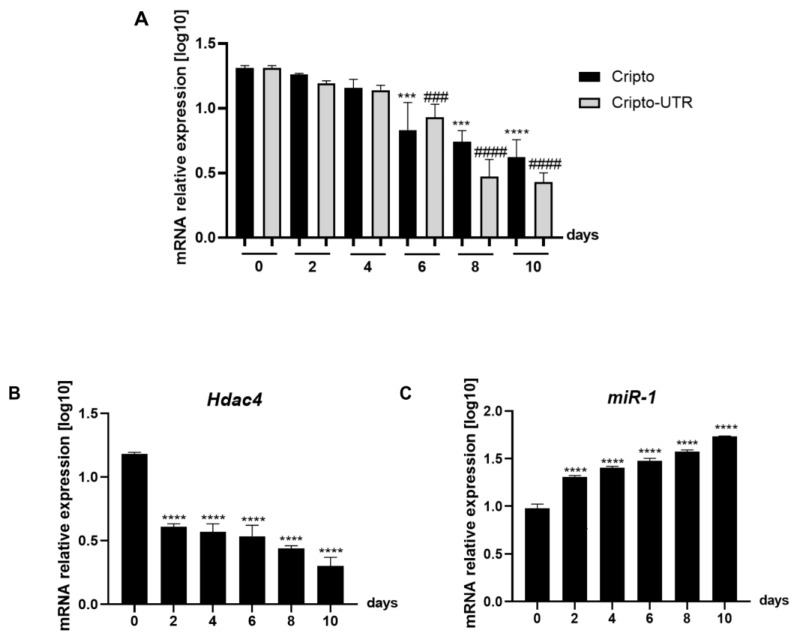
The expression level of *Cripto, Cripto-UTR, Hdac-4*, and miRNA-1 in mouse P19 cells undergoing cardiac differentiation. (**A**–**C**) qPCR analysis in P19 cells during cardiomyogenesis. The mRNA levels of *Cripto*, *Cripto-UTR*, and *Hdac-4* were normalised to *Gapdh* levels, while the levels of miR-1 were normalised to U6 levels. The data are expressed as the means (SDs). The significance was determined by one-way ANOVA, followed by Dunnett’s multiple comparison test. (**A**) *** (*p* < 0.001) and **** (*p* < 0.0001) represent the significance of Cripto vs. day 0 (undifferentiated cells); ^###^ (*p* < 0.001) and ^####^ (*p* < 0.0001) represent the significance of Cripto-UTR vs. day 0 (undifferentiated cells). (**B**) **** (*p* < 0.0001) represents the significance of Hdac-4 vs. day 0 (undifferentiated cells) differentiation. (**C**) **** (*p* < 0.0001) represents the significance of miR-1 vs. day 0 (undifferentiated cells).

**Figure 6 ijms-24-12251-f006:**
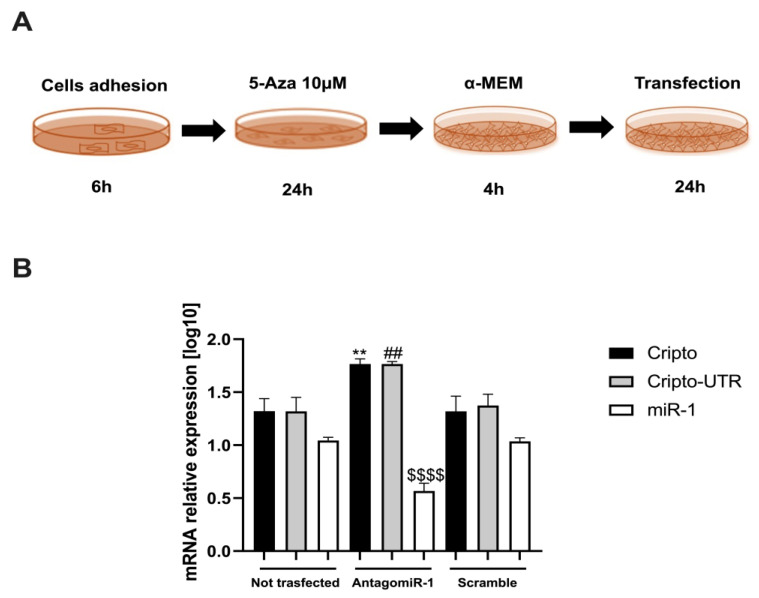
AntagomiR-1 gene transfection. (**A**) Schematic representation of the experimental protocol used for AntagomiR-1 transfection in P19 cells. (**B**) qPCR analysis in P19 cells after AntagomiR-1 transfection. The mRNA levels of *Cripto* and *Cripto-UTR* were normalised to *Gapdh* levels, while the levels of miR-1 were normalised to U6 levels. The data are expressed as the means (SDs). The significance was determined by one-way ANOVA, followed by Dunnett’s multiple comparison test. ** (*p* < 0.01) represents the significance of *Cripto* vs. Not transfected; ^##^ (*p* < 0.01) represents the significance of *Cripto-UTR* vs. Not transfected; ^$$$$^ (*p* < 0.0001) represents the significance of miR-1 vs. Not transfected.

**Figure 7 ijms-24-12251-f007:**
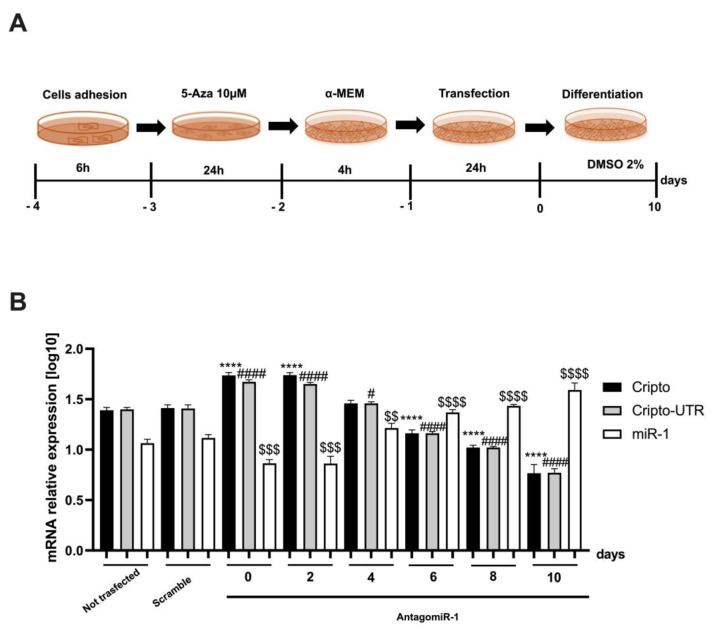
AntagomiR-1 gene transfection before P19 cardiac differentiation. (**A**) Schematic representation of the experimental protocol used for AntagomiR-1 transfection in P19 cells undergoing cardiac differentiation. (**B**) qPCR analysis of P19 cells after AntagomiR-1 transfection. The mRNA levels of *Cripto* and *Cripto-UTR* were normalized to *Gapdh* levels, while the levels of miR-1 were normalized to U6 levels. The data are expressed as the means (SDs). The significance was determined by one-way ANOVA, followed by Dunnett’s multiple comparison test. **** (*p* < 0.0001) represents the significance of *Cripto* vs. Not transfected; ^#^ (*p* < 0.05) and ^####^ (*p* < 0.0001) represent the significance of *Cripto-UTR* vs. Not transfected; ^$$^ (*p* < 0.01), ^$$$^ (*p* < 0.001), and ^$$$$^ (*p* < 0.0001) represent the significance of miR-1 vs. Not transfected.

**Figure 8 ijms-24-12251-f008:**
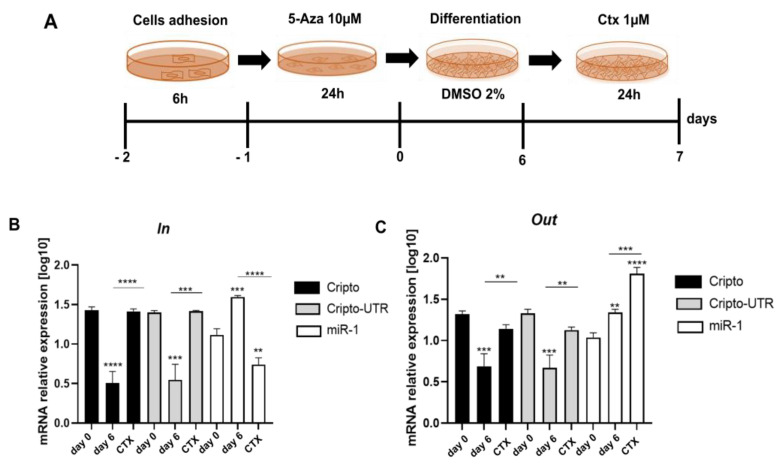
Gene expression analysis of miR-1 and Cripto after CTX-damage in P19 cells differentiated in cardiomyocytes. (**A**) Schematic representation of the experimental protocol used. (**B**,**C**) qPCR analysis of P19 cells. The mRNA levels of Cripto and Cripto-UTR were normalized to *Gapdh* levels, while the levels of miR-1 were normalized to U6 levels. The data are expressed as the means (SDs). The significance was determined by one-way ANOVA followed by Dunnett’s multiple comparison test. (**B**) Gene expression analysis inside the cells. ** (*p* < 0.01), *** (*p* < 0.001), and **** (*p* < 0.0001). (**C**) Gene expression analysis outside (culture medium) the cells. ** (*p* < 0.01), *** (*p* < 0.001), and **** (*p* < 0.0001).

**Figure 9 ijms-24-12251-f009:**
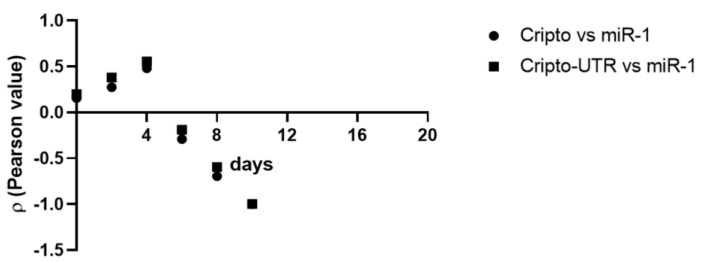
Correlation between Cripto and miR-1 during cardiac differentiation. Values obtained from the Pearson’s analysis are reported on the *y*-axis; days are reported on the *x*-axis.

**Figure 10 ijms-24-12251-f010:**
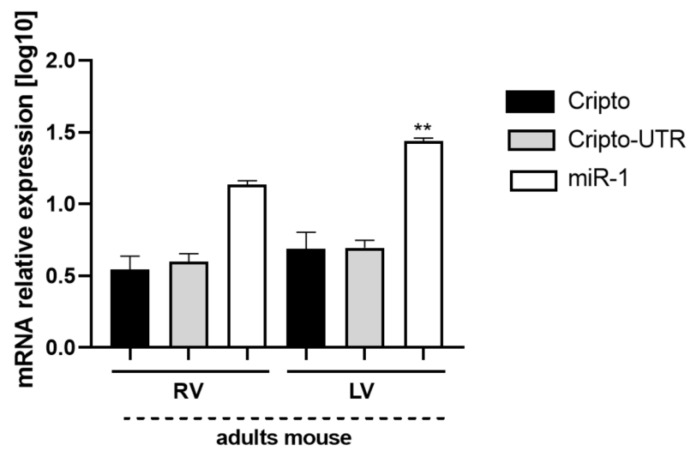
Expression of miR-1 and *Cripto* at a ventricular level in *mouse heart* ex vivo tissues. The qPCR analysis of ex vivo tissues of adults mouse hearts; six mouse hearts were used for each point shown in the graph. The mRNA levels of *Cripto* and *Cripto-UTR* were normalised to *Gapdh* levels, while the levels of miR-1 were normalised to U6 levels. The data are expressed as the means (SDs). The significance was determined by one-way ANOVA, followed by Dunnett’s multiple comparison test. Gene expression analysis inside the cells. ** (*p* < 0.01).

**Table 1 ijms-24-12251-t001:** Pearson’s correlation (ρ) between Cripto, Cripto-UTR, and miR-1.

Variables			ρ			
Days						
	0	2	4	6	8	10
Cripto vs. miR-1	0.16	0.27	0.48	−0.29	−0.70	−0.99
Cripto-UTR vs. miR-1	0.20	0.38	0.56	−0.19	−0.60	−0.99

**Table 2 ijms-24-12251-t002:** List of genes and sequences of primers used for qPCR analysis.

Gene	Primer for 5′-3′	Primer rev 5′-3′
*Gapdh*	GGTGAAGGTCGGTGTGAACG	CTCGCTCCTGGAAGATGGTG
*Cripto-utr*	GACAGACAGGCCTACACAGA	TCGCTACATAGACCAGGCTG
*Cripto*	TGGACGCAACTGTGAACATG	TTGAGGTCCTGGTCCATCAC
*Nkx 2.5*	CAGAACCGTCGCTACAAGTG	GGTAGGGGTAGGCGTTGTAG
*Gata-4*	GTTACCTGGCTCTGGGACTT	GTGGGTGATGAGGACAAGGA
*Apj*	CCAGTGTCTTTTGCCTCACC	CTGAGTTTGAAGTGGCCACC
*Mlc-2*	ATCAAAGAGGCTCCAGGTCC	GTCAGCATCTCCCGGACATA

## Data Availability

Not applicable.

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
