# Peer review of "Cripto Is Targeted by miR-1a-3p in a Mouse Model of Heart Development"

_ijms, 2023, doi:10.3390/ijms241512251_

Round 1

Reviewer 1 Report

This manuscript was well written overall and is a good investigation of the cross talk between criptio and myomirR-1a in cardiomyocyte differentiation.

There a few concerns:

1.   To address the specificity of miR1a, a control should be tested where the specific binding site is mutated instead of the scrambled control which was not well described.

2. Figure 3, western blot for Cripto would strengthen this section.

3.   For P19 differentiation, functional aspects such as beating foci of cells or calcium currents should be assessed to demonstrate their differentiation functionally.

4.   Line 182, 5-azacytidine is not a specific tool and that should be addressed in the manuscript.

5.  Figure 5 legend  should be re-written with respect to the statistics.  Currently it says "represents the significance of Hadc-4 during cardiac differentiation"   more accurately this is simiply the difference from the undifferentiated cells and should be phrased that way.

6.  The whole heart tissue was not as well developed as the rest of the studies.   Only postnatal tissues are looked at whereas most of cardiac differentiation has occurred prenatally.   Embryonic tissues should be used as well.   In the whole hearts, it was not clear what tissues where expressing Cripto.   Immunohistochemistry or cell isolation would strengthen this part.

The writing was well done. No concerns.

Author Response

Dear Editor,

thank you for the Report about our manuscript entitled “Cross-talking between Cripto and myomiR-1a-3p during cardiac differentiation in a mouse model system”, submitted to IJMS. We appreciate your comments and have carefully re-considered them in preparing a new manuscript version.

A point-by-point response to the comments is attached below.

We believe that the manuscript has now greatly improved thanks to the input received.

We hope that the new version of the paper deserves publication on IJMS.

Best regards,

Dr. Mariarita Brancaccio and Prof. Tiziana Angrisano

Point-by-point response.

Reviewer 1

Comments and Suggestions for Authors

This manuscript was well written overall and is a good investigation of the cross talk between cripto and myomirR-1a in cardiomyocyte differentiation.

There a few concerns:

  1. To address the specificity of miR1a, a control should be tested where the specific binding site is mutated instead of the scrambled control which was not well described.

Response

Thanks for the comments.

To establish Cripto as the direct target of miR1 we performed a transfection in HEK-293 cells with the anti-miR-1 Antisense Inhibitor (Antagomir-1), as described in the materials and methods session, paragraph 4.2.

  1. Figure 3, western blot for Cripto would strengthen this section.

Response

Thanks for the comments.

In accordance with your observations, we have added Figure 3C where the western blot for Cripto can be observed, in the Results session, paragraph 3.2.

  1. For P19 differentiation, functional aspects such as beating foci of cells or calcium currents should be assessed to demonstrate their differentiation functionally.

Response

Thanks for the comments.

To verify that the differentiation has taken place correctly and that the cells are cardiomyocytes, we used the analysis of the protein expression levels of cardiac troponin, a specific marker as reported in figures 4A and 4B and in references 8 and 9 (Gomes, A.V. et al., The role of troponins in muscle contraction. IUBMB Life. 2002, 6, 323-33; 9.    Sharma, S.; Jackson, P.G.; Makan, J. Cardiac troponins. J Clin Pathol. 2004, 10, 1025-6).

In fact, the presence of Troponin, as can be seen from the densitometric analysis, is not significant between days 0 and 4; but it turns out to be from day 6; only then when we have a "maturation of the cells towards cardiac differentiation" in accordance with what is described in the literature.

  1. Line 182, 5-azacytidine is not a specific tool and that should be addressed in the manuscript.

Response

Thanks for the comments.

Azacytidine appears to have a fundamental role in cardiac differentiation as reported by Qian, Q. et al., 5-Azacytidine induces cardiac differentiation of human umbilical cord-derived mesenchymal stem cells by activating ex-tracellular regulated kinase. Stem Cells Dev. 2012, 1, 67-75 and Makino, S. et al., Cardiomyocytes can be generated from marrow stromal cells in vitro. J Clin Invest. 1999, 5, 697-705.

We have therefore reported these references in text n. 36 and 37 respectively.

  1. Figure 5 legend should be re-written with respect to the statistics.  Currently it says "represents the significance of Hadc-4 during cardiac differentiation"   more accurately this is simiply the difference from the undifferentiated cells and should be phrased that way.

Response

Thanks for the comments.

In accordance with your comment, we have modified the legend of Figure 5.

  1. The whole heart tissue was not as well developed as the rest of the studies. Only postnatal tissues are looked at whereas most of cardiac differentiation has occurred prenatally.   Embryonic tissues should be used as well.   In the whole hearts, it was not clear what tissues where expressing Cripto.   Immunohistochemistry or cell isolation would strengthen this part.

Response

Thanks for the comments.

A clarification regarding the use of ex vivo murine tissues is necessary. In fact, it is known that morphology and cardiovascular development between humans and mice are comparable (Krishnan A, Samtani R, Dhanantwari P, Lee E, Yamada S, Shiota K, Donofrio MT, Leatherbury L, Lo CW. A detailed comparison of mouse and human cardiac development. Pediatr Res. 2014 Dec;76(6):500-7. doi: 10.1038/pr.2014.128. Epub 2014 Aug 28. PMID: 25167202; PMCID: PMC4233008). Therefore, we decided to use mice at different days and use the whole heart to understand if Cripto and/or miR-1 could be used as cardiac markers. In fact, surely our future studies will be to monitor their levels in serum (so as to make them easily accessible markers and above all monitorable in a short time) as cardiac biopsies are invasive methods.

Comments on the Quality of English Language

The writing was well done. No concerns

We hope that the changes made are satisfactory, thanks for the support.

All modification was highlight in yellow in the text.

Reviewer 2

Comments and Suggestions for Authors

Dear authors of the work ijms-2469940, allow me to make some constructive observations on your work.

Aim

In the abstract you mention that your objective was to make an inverse correlation between Crip and miR-1, however, in the methodologies you do not include any statistical test to measure the inverse correlation.

Also, in the introduction section you mention that your objective was to evaluate a cross-talk between the Cripto and miR-1.

1     Please clearly define the objective of the work.

Thanks for the comments.

In accordance with your comment, we performed a new analysis. In particular, to confirm the inverse relationship between Cripto and miR-1 we performed Pearson's statistical test (Akoglu H. User's guide to correlation coefficients. Turk J Emerg Med. 2018 Aug 7;18(3):91-93. doi: 10.1016/j.tjem.2018.08.001. PMID: 30191186; PMCID: PMC6107969.). The Pearson’s test was described: in the materials and methods session, in 4.13 paragraph, pages 17 from 586 to 595 lines, we explained the test and in the results in 3.8 paragraph, pages 12-13 from 323 to 349 lines, we reported the data obtained.

Introduction

2 The idea that begins on line 43 and ends at the beginning of line 46 must be cited.

Thanks for the comments.

In accordance with your comment, we have added reference number 3 (Meilhac SM, Lescroart F, Blanpain C, Buckingham ME. Cardiac cell lineages that form the heart. Cold Spring Harb Perspect Med. 2014 Sep 2;4(9):a013888. doi: 10.1101/cshperspect.a013888) (page 2, line 50).

3 The description of the Cripto protein found between lines 63-69 is an unacceptable plagiarism of an article published on Wikipedia, I attach the link : https://en.wikipedia.org/wiki/Cripto#cite_note-CFC1-8.

Thanks for the comments.

In accordance with your comment, we have modified the Cripto description as reported at lines 66-76.

Results

Figure 2, 3B, 4D, 4E, 4F, 4G, 5A, 5B, 5C, 6B, 7B, 8B, 8C, 9A, 9B, 9C and 10

In the descriptions of all the figures you mention that the data is represented as the means +/- SDs . However, in the bars only the SDs (+) are visible, and the value (-) is omitted. It must be remembered that the value of any SDs is the square root of the variance and all the results of a square root are two numbers: one positive and one negative.

  1. Please include the values (-) of the SDs

Thanks for the comments.

In accordance with your comment, we have modified the legend and the statistical analysis and in addition, we have added reference number 71 (Jaykaran. "Mean ± SEM" or "Mean (SD)"? Indian J Pharmacol. 2010 Oct;42(5):329. doi: 10.4103/0253-7613.70402. PMID: 21206631; PMCID: PMC2959222).

Discussion

In general, the discussion is poor, since they abundantly describe all the results again and rarely contrast (discuss) the results.

Line

5     315               Again describe the objective

6     315-334        Basically they repeated the results obtained

7    335               The idea must be cited

8     336-337        The idea should be cited

9    337-339        The idea should be cited

10   366               It is suggested to change “the results in our possession”

11   380               It is suggested to change “Cripto and miR-1 “ talk ””

12  397-400        The idea is not well posed, since you did NOT do None proof on acute myocardial infarction of the ST-segment. Please rephrase that part of the discussion .

Thanks for the comments.

In accordance with your comment, we have modified the first part of the discussion, in section 3, and added two new references where required. In addition, we changed “the results in our possession” to “our data” and “talk” to “related”. Finally, we have rephrased that part of the discussion line 397-400 now line 428-433.

13 Conclusion: They usually conclude something they didn't: a statistical test of inverse correlation.

Thanks for the comments.

In accordance with your comment, we performed a new analysis. In particular, to confirm the inverse relationship between Cripto and miR-1 we performed Pearson's statistical test (Akoglu H. User's guide to correlation coefficients. Turk J Emerg Med. 2018 Aug 7;18(3):91-93. doi: 10.1016/j.tjem.2018.08.001. PMID: 30191186; PMCID: PMC6107969.)The Pearson’s test was described: in the materials and methods session, in 4.13 paragraph, pages 17 from 586 to 595 lines, we explained the test and in the results in 3.8 paragraph, pages 12-13 from 323 to 349 lines, we reported the data obtained.

14 In the conclusions the results are NOT discussed, please move the ideas between lines 531-533 to the discussion section.

Response

Thanks for the comments.

In accordance with your comment, we have modified the conclusions paragraph 5 in order not to be repetitive.

We hope that the changes made are satisfactory, thanks for the support.

All modification was highlight in yellow in the text.

Reviewer 2 Report

Dear authors of the work ijms-2469940, allow me to make some constructive observations on your work.

Aim

In the abstract you mention that your objective was to make an inverse correlation between Crip and miR-1, however, in the methodologies you do not include any statistical test to measure the inverse correlation.

Also, in the introduction section you mention that your objective was to evaluate a cross-talk between the Crip and miR-1.

1     Please clearly define the objective of the work.

Introduction

2 The idea that begins on line 43 and ends at the beginning of line 46 must be cited.

3 The description of the Cripto protein found between lines 63-69 is an unacceptable plagiarism of an article published on Wikipedia, I attach the link : https://en.wikipedia.org/wiki/Cripto#cite_note-CFC1-8.

Results

Figure 2, 3B, 4D, 4E, 4F, 4G, 5A, 5B, 5C, 6B, 7B, 8B, 8C, 9A, 9B, 9C and 10

In the descriptions of all the figures you mention that the data is represented as the means +/- SDs . However, in the bars only the SDs (+) are visible, and the value (-) is omitted. It must be remembered that the value of any SDs is the square root of the variance and all the results of a square root are two numbers: one positive and one negative.

4.     Please include the values (-) of the SDs

Discussion

In general, the discussion is poor, since they abundantly describe all the results again and rarely contrast (discuss) the results.

Line

5     315               Again describe the objective

6     315-334        Basically they repeated the results obtained

  335               The idea must be cited

8     336-337        The idea should be cited

9    337-339        The idea should be cited

 366               It is suggested to change “the results in our possession”

 380               It is suggested to change “Cripto and miR-1 “ talk ””

1  397-400        The idea is not well posed, since you did NOT do None proof on acute myocardial infarction of the ST-segment. Please rephrase that part of the discussion .

Conclusion

They usually conclude something they didn't: a statistical test of inverse correlation.

In the conclusions the results are NOT discussed, please move the ideas between lines 531-533 to the discussion section.

Author Response

(The authors gave the same response as above.)

Round 2

Reviewer 1 Report

The authors have partially responded to my concerns but the main issues remain especially the studies with heart tissue remain.

"The studies on the whole heart tissue was not as well developed as the rest of the studies. Only postnatal tissues are looked at whereas most of cardiac differentiation has occurred prenatally.   Embryonic tissues should be used as well.   In the whole hearts, it was not clear what tissues where expressing Cripto.   Immunohistochemistry or cell isolation would strengthen this part."

As well, the control for luciferase with the binding site mutated should be included.

Minor points:

1.   Line 22 "developing heart's myocardium" to "developing myocardium"

2.  At the end of the introduction roman numerals were not used properly.  i, ii, iii, iv,  v, vi   not i, ii, iii, iiii, iiiii etc...

Author Response

Dear Editor,

thank you for the Report about our manuscript entitled “Cross-talking between Cripto and myomiR-1a-3p during cardiac differentiation in a mouse model system”, submitted to IJMS. We appreciate your comments and have carefully re-considered them in preparing a new manuscript version.

A point-by-point response to the comments is attached below.

We believe that the manuscript has now greatly improved thanks to the input received.

We hope that the new version of the paper deserves publication on IJMS.

Best regards,

Dr. Mariarita Brancaccio and Prof. Tiziana Angrisano

Point-by-point response.

Academic Editor

"As indicated by reviewer 1 it is important to show the correlation between Cripto and myomiR-1a-3p at earlier stages of development when cardiomyocyte differentiation is occuring. I suggest performing the same qPCR analysis as in Fig.9. but include some earlier stages to show that the correlation between Cripto and myomiR-1a-3p is relevant during this process."

Response

Thanks for your observation, I'm sorry if I was not clear in writing and exposing past and current findings regarding the role of Cripto and myomiR-1a-3p (miR-1).

In this case, Minchiotti, author of this manuscript too, together with Parisi, demonstrated the role of Cripto in cardiomyogenesis (see references 10 and 12 inside the manuscript); concurrently, myomiR-1a-3p (miR-1) as reported in references 24 and 25 of the manuscript is a major player involved in proliferation, differentiation and cardiac disorders.

Thus, the reason why we used mouse hearts at different days and not mouse hearts during embryonic development was because both were already known to be involved.

In our work, we have recreated an in vitro model of the heart, of course, we have carried out cardiac differentiation monitored by the expression of known regulatory markers of this differentiation; however the ultimate goal was to use this model as a valid tool to search for new therapeutic targets.

In this case, first with a bioinformatic and subsequently molecular approach, we have shown that Cripto is a target of miR-1 (data not known to date) and that this data could be useful for exploiting both as rapid access serum markers and possibly therapeutic targets (being the circulating microRNAs), since cardiac biopsies are invasive and very often are performed when the damage is already extensive and irreversible.

In particular, the experiment with ctx and in and out monitoring was to mimic how it happens during heart damage.

The whole heart was therefore needed to understand that they were both expressed and that there was a correlation even as the days increased. So they wanted valid biomarkers that could possibly be exploited as serum markers (future experiments on which we will focus).

In fact, our future studies will focus on identifying biomolecules useful for regulating both during cardiac damage since numerous scientific works highlight the new frontier of biomarkers in microRNAs.

The title was probably already misleading; for this purpose, we changed the current title: “Cross-talking between Cripto and myomiR-1a-3p during cardiac differentiation in a mouse model system” to Cripto is targeted by miR-1a-3p in a mouse model of heart. In addition, we have appointed some changes in the text to elucidate our aim.

Thanks for the support, we hope we have been more explanatory in clarifying the purpose of our work and the data reported.

All modification was highlighted in green in the text.

Reviewer 1

Comments and Suggestions for Authors

The authors have partially responded to my concerns but the main issues remain especially the studies with heart tissue remain.

"The studies on the whole heart tissue was not as well developed as the rest of the studies. Only postnatal tissues are looked at whereas most of cardiac differentiation has occurred prenatally.   Embryonic tissues should be used as well.   In the whole hearts, it was not clear what tissues where expressing Cripto.   Immunohistochemistry or cell isolation would strengthen this part."

As well, the control for luciferase with the binding site mutated should be included.

Response

Thanks for your observation, I'm sorry if I was not clear in writing and exposing past and current findings regarding the role of Cripto and myomiR-1a-3p (miR-1).

In this case, Minchiotti, author of this manuscript too, together with Parisi, demonstrated the role of Cripto in cardiomyogenesis (see references 10 and 12 inside the manuscript); concurrently, myomiR-1a-3p (miR-1) as reported in references 24 and 25 of the manuscript is a major player involved in proliferation, differentiation and cardiac disorders.

Thus, the reason why we used mouse hearts at different days and not mouse hearts during embryonic development was because both were already known to be involved.

In our work, we have recreated an in vitro model of the heart, of course, we have carried out cardiac differentiation monitored by the expression of known regulatory markers of this differentiation; however the ultimate goal was to use this model as a valid tool to search for new therapeutic targets.

In this case, first with a bioinformatic and subsequently molecular approach, we have shown that Cripto is a target of miR-1 (data not known to date) and that this data could be useful for exploiting both as rapid access serum markers and possibly therapeutic targets (being the circulating microRNAs), since cardiac biopsies are invasive and very often are performed when the damage is already extensive and irreversible.

In particular, the experiment with ctx and in and out monitoring was to mimic how it happens during heart damage.

The whole heart was therefore needed to understand that they were both expressed and that there was a correlation even as the days increased. So, they wanted valid biomarkers that could possibly be exploited as serum markers (future experiments on which we will focus).

In fact, our future studies will focus on identifying biomolecules useful for regulating both during cardiac damage since numerous scientific works highlight the new frontier of biomarkers in microRNAs.

The title was probably already misleading; for this purpose, we changed the current title: “Cross-talking between Cripto and myomiR-1a-3p during cardiac differentiation in a mouse model system” to Cripto is targeted by miR-1a-3p in a mouse model of heart. In addition, we have appointed some changes in the text to elucidate our aim.

The Luciferase assay and the controls used were performed in accordance with what was previously described in the work of Pan et al. (reference 65, cited in the materials and methods)

Minor points:

  1. Line 22 "developing heart's myocardium" to "developing myocardium"

Response

Thanks for the comments, in accordance with your suggestion, we changed the text.

  1. At the end of the introduction roman numerals were not used properly. i, ii, iii, iv,  v, vi   not i, ii, iii, iiii, iiiii etc...

Response

Thanks for the comments, in accordance with your suggestion, we changed the text.

Thanks for the support, we hope we have been more explanatory in clarifying the purpose of our work and the data reported.

All modification was highlighted in green in the text.

Reviewer 2

Comments and Suggestions for Authors

Dear authors of the ijms-2469940 paper, I want to express my congratulations for the improvements to your research paper. Sincerely.

Thank you very much for your support and suggestions, we are happy to have satisfied your requests.

Reviewer 2 Report

Estimados autores del trabajo ijms-2469940, quiero expresar mis felicitaciones por las mejoras a su trabajo de investigación. Atentamente.

Dear authors of the ijms-2469940 paper, I want to express my congratulations for the improvements to your research paper. Sincerely.

Author Response

(The authors gave the same response as above.)

Round 3

Reviewer 1 Report

The new title is incomplete as written.  "Cripto is targeted by miR-1a-3p in a mouse model of heart"    Model of heart development??

The in vivo data is not well developed or rationalized as the timepoints in development in the embryoid  bodies and P19 cells are much earlier than in postnatal cells.   My suggestion would be to remove the two in vivo figures as it does not add to the article but distracts from it.

Author Response

Dear Editor,

thank you for the Report about our manuscript entitled “Cross-talking between Cripto and myomiR-1a-3p during cardiac differentiation in a mouse model system”, submitted to IJMS. We appreciate your comments and have carefully re-considered them in preparing a new manuscript version.

A point-by-point response to the comments is attached below.

We believe that the manuscript has now greatly improved thanks to the input received.

We hope that the new version of the paper deserves publication on IJMS.

Best regards,

Dr. Mariarita Brancaccio and Prof. Tiziana Angrisano

Point-by-point response.

Reviewer 1

Comments and Suggestions for Authors

The new title is incomplete as written.  "Cripto is targeted by miR-1a-3p in a mouse model of heart"    Model of heart development??

The in vivo data is not well developed or rationalized as the timepoints in development in the embryoid  bodies and P19 cells are much earlier than in postnatal cells.   My suggestion would be to remove the two in vivo figures as it does not add to the article but distracts from it.

Response

In accordance with your suggestions we have changed the title and eliminated the figure in which there were mouse hearts from day 1 to adult.

We exclusively kept the figure in which we evaluate Cripto and miR-1 gene expression in adult heart biopsies to confirm the intent to use these as cardiac biomarkers.

As we have reported both in the discussion and in the conclusions.

We hope that the text is now to your liking, thanks for the support

All modification was highlighted in light green in the text.

Round 4

Reviewer 1 Report

No further changes.